Strain- and plasmid-level deconvolution of a synthetic metagenome by sequencing proximity ligation products

Beitel Christopher W. 1 cb@ucdavis.edu
Froenicke Lutz 1
Lang Jenna M. 1
Korf Ian F. 1 2
Michelmore Richard W. 1 2 3
Eisen Jonathan A. 1 4 5
Darling Aaron E. 6 aaron.darling@uts.edu.au
1 The University of California, Davis Genome Center , Davis, CA , USA
2 Department of Molecular and Cellular Biology, University of California , Davis, CA , USA
3 Department of Plant Sciences, University of California , Davis, CA , USA
4 Department of Medical Microbiology and Immunology, University of California , Davis, CA , USA
5 Department of Evolution and Ecology, University of California , Davis, CA , USA
6 ithree institute, University of Technology Sydney , Sydney, NSW , Australia
Crandall Keith
Electronic publication date: 2014 May 27
Publication date: 2014
Volume: 2
Electronic Location ID: e415
Received 2014 Mar 3; Accepted 2014 May 15
Copyright: © 2014 Beitel et al.
Copyright year: 2014
Copyright holder: Beitel et al.
License: This is an open access article distributed under the terms of the Creative Commons Attribution License, which permits unrestricted use, distribution, reproduction and adaptation in any medium and for any purpose provided that it is properly attributed. For attribution, the original author(s), title, publication source (PeerJ) and either DOI or URL of the article must be cited.
License URL: https://creativecommons.org/licenses/by/4.0/

Keywords: Hi-C, Microbial ecology, Metagenomics, Plasmids, Synthetic microbial communities, Markov clustering, Metagenome assembly, Strain differentiation, Haplotype phasing, Genome scaffolding

Funding: MARS, Inc Department of Homeland Security contract HSHQDC-11-C-00091 This work was supported by a gift from MARS, Inc. and by Department of Homeland Security contract #HSHQDC-11-C-00091. The funders had no role in study design, data collection and analysis, decision to publish, or preparation of the manuscript.

==============================
Metagenomics is a valuable tool for the study of microbial communities but has been limited by the difficulty of “binning” the resulting sequences into groups corresponding to the individual species and strains that constitute the community. Moreover, there are presently no methods to track the flow of mobile DNA elements such as plasmids through communities or to determine which of these are co-localized within the same cell. We address these limitations by applying Hi-C, a technology originally designed for the study of three-dimensional genome structure in eukaryotes, to measure the cellular co-localization of DNA sequences. We leveraged Hi-C data generated from a simple synthetic metagenome sample to accurately cluster metagenome assembly contigs into groups that contain nearly complete genomes of each species. The Hi-C data also reliably associated plasmids with the chromosomes of their host and with each other. We further demonstrated that Hi-C data provides a long-range signal of strain-specific genotypes, indicating such data may be useful for high-resolution genotyping of microbial populations. Our work demonstrates that Hi-C sequencing data provide valuable information for metagenome analyses that are not currently obtainable by other methods. This metagenomic Hi-C method could facilitate future studies of the fine-scale population structure of microbes, as well as studies of how antibiotic resistance plasmids (or other genetic elements) mobilize in microbial communities. The method is not limited to microbiology; the genetic architecture of other heterogeneous populations of cells could also be studied with this technique.

Introduction

Microbial ecology is the study of microbial communities in terms of their composition, functional diversity, interactions, stability, and emergent properties (Handelsman, 2004; Konopka, 2009). Knowledge of the roles microbes play in ecosystems is essential for understanding how these ecosystems function (Konopka, 2009). Readily-cultivated organisms are estimated to constitute less than 1% of all microbial species, leading to the development of culture-independent methods for studying microbial communities (Gilbert & Dupont, 2011; Hugenholtz, 2002; Staley, 1985). These culture-independent methods allow communities to be characterized directly.

Current sequencing-based metagenomic methods do not capture some of the most informative genetic information in microbial communities, in particular the long-range sequence contiguity and associations of genetic material in individual cells. In nearly all metagenomic methods, cells from the microbial community are lysed en masse to obtain a bulk DNA sample. This results in DNA from many different cells being mixed together, so that the genotype and species identity of individual cells are lost. Chromosomal DNA is then fragmented into pieces (∼500 bp–40 kbp, depending on the sequencing strategy), further reducing contiguity.

Improved sample-processing workflows might preserve this information and thereby yield greater insight into the genetic structure of microbial communities. High throughput single-cell genomics (e.g., applied to thousands of cells) offers a promising alternative to shotgun metagenomics that preserves information about cellular compartmentalization of genetic material. These approaches are exquisitely sensitive to contamination by foreign DNA (from the sample itself, the laboratory environment, and from “ultra-pure” commercial reagents), necessitating the use of specialized equipment and reagents (Blainey, 2013; Woyke et al., 2011). Long-read technologies, such as Pacific Biosciences (Eid et al., 2009) and nanopore (Maitra, Kim & Dunbar, 2012) sequencing, may help address this challenge but are still constrained by the difficulty of preparing adequate amounts of very long DNA fragments.

Computational methods have been developed to infer genomic contiguity from metagenomic data by binning metagenome assembly contigs by species. These binning procedures pose a significant analytical challenge. Several methods have been developed that can be divided into comparative, compositional, and assembly approaches. Comparative approaches use alignments to reference sequences to assign contigs to species within existing taxonomies (Droge & McHardy, 2012). Comparative approaches are limited by their reliance on existing taxonomies. Compositional approaches form clusters of contigs that share similar oligomer (usually 4 bp–8 bp) composition (Droge & McHardy, 2012). Compositional approaches tend to be limited as well due to their underlying assumption that contigs with similar sequence composition belong together. Horizontal gene transfer complicates both of these analysis methods because it can introduce gene content from a taxonomically distant relative with unusual nucleotide composition. Thirdly, metagenome assembly can be viewed as a metagenome binning approach since sequences placed on the same scaffold are necessarily present in the same bin for any downstream binning procedure. As with all genome assembly approaches, metagenome assembly seeks to infer sequence adjacencies from paired- and long-read technologies. This approach is limited by the availability of such data that span large repetitive regions (Treangen & Salzberg, 2012; Howe et al., 2014; Treangen et al., 2013).

Binning seeks to address the challenge of determining which sequences were present within cells of the same species prior to DNA extraction. We reasoned that such co-localizations could be inferred from Hi-C data, a method originally developed for the study of three-dimensional genome structure in eukaryotes (Lieberman-Aiden et al., 2009). This method relies on cross-linking molecules in close physical proximity and consequently identifies both intra- and inter-chromosomal associations, reflecting the spatial arrangement of DNA at the time of cross-linking within intact nuclei or non-nucleated cells (Umbarger et al., 2011). We predicted that sequences of DNA not present in the same cell at the time of cross-linking would not be cross-linked together and should not be associated by Hi-C reads (Fig. S1). Herein we demonstrate the utility of Hi-C as a tool for addressing metagenomic binning and related problems in microbial ecology. To do so we first constructed a synthetic microbial community by culturing and mixing five organisms with available reference genomes. We then performed a metagenome assembly on sequences that were simulated in silico from the genomes of these organisms. Our first objective was to group these metagenome assembly contigs according to species using Hi-C reads that were generated from the synthetic microbial community. We then sought to differentiate two closely related E. coli strains included within this mixture. To do so we constructed contig and variant graphs and analyzed those graphs to characterize the extent to which Hi-C data might resolve the genotypes of species and strains present in our synthetic community.

Materials and Methods

Construction of a synthetic microbial community

Pediococcus pentosaceus and Lactobacillus brevis were provided by the UC Davis Enology Culture Collection (http://wineserver.ucdavis.edu). Single colonies were used to start cultures in 5 ml liquid MRS broth. Escherichia coli BL21 (ATCC# PTA-5073), E. coli K12 DH10B (ATCC# 207214), and Burkholderia thailandensis (ATCC# 700388) were obtained as freeze-dried stocks from the American Type Culture Collection (ATCC). The E. coli strains were re-suspended in 5 ml of LB liquid medium (10 g/L Tryptone, 10 g/L NaCl, 5 g/L Yeast Extract) and the B. thailandensis was re-suspended in 5 ml of Nutrient Broth (Peptone 15.0 g/L, yeast extract 3 g/L, sodium chloride 6 g/L, D(+)glucose 1 g/L). All were incubated, with shaking, overnight at 37 °C to produce starter cultures.

A separate 50 ml culture for each organism was created by inoculation with 10 µl of the appropriate starter culture and grown, with shaking, at 37 °C, for 24 h. The cell density of each culture was estimated by measuring the OD600. The cultures were then mixed in quantities proportional to their optical density, seeking to have equal representation of each organism in the synthetic community. Glycerol was added to a final concentration of 7% and the final mixture was divided into 2 ml tubes and frozen at −80 °C.

Simulated metagenome assemblies

We simulated Illumina paired-end sequencing of the synthetic microbial community to obtain metagenomic assemblies that we could subsequently attempt to bin using experimentally derived Hi-C reads. Reads were simulated using Grinder (Angly et al., 2012) v0.4.5, a tool for simulating metagenomic shotgun sequence reads. Replicons were sampled assuming uniform abundance of species. A total of 61,063,000 reads were simulated to cover the genomes at 500x. From this set, paired-end read datasets of varying levels of coverage (100, 50, and 5x) were formed. Read length was simulated to 165 bp and fragment size was simulated with a normal distribution around 550 bp with a standard deviation of 50 bp (grinder -am uniform -cf $cov-rd 165 -id 550 normal 50 -rf $ref -fq 1 -ql 30 10 -bn grinder.dp$cov.$tag).

Assembly from each of the simulated metagenomic read sets was performed using SOAPdenovo (Luo et al., 2012) with a k-mer length of 23, yielding assemblies of varying quality (Table S1). Reads were aligned to the resulting assembly contigs using BWA MEM (Li, 2013). The rate of misassembly was determined by mapping contigs back to the reference assemblies (BWA-MEM with default parameters and a quality filter of MapQ > 20) and counting the number of contigs that joined sequences from different species. These assemblies are available on Figshare: http://dx.doi.org/10.6084/m9.figshare.1004473, http://dx.doi.org/10.6084/m9.figshare.1004472, http://dx.doi.org/10.6084/m9.figshare.1004471.

Application of Hi-C to the synthetic microbial community

We performed Hi-C on the synthetic microbial community for the purpose of obtaining information that could be used to group (by species) the simulation-derived contigs described above, as well as to differentiate closely related strains present in our synthetic community. Hi-C was carried out by combining the cross-linking and cell wall digestion procedures described by Umbarger et al. (2011) for bacterial 3-C experiments and the Hi-C protocol developed for mammalian cells (Lieberman-Aiden et al., 2009) with minor modifications. For additional details, see Supplemental Information: Hi-C of Mixed Bacterial Cultures. Cells were transferred into a 50 ml centrifuge tube and washed three times in 25 ml of TE buffer (pH = 8.0) by centrifugation for 5 min at 4000 rpm at 4 °C. Cells were re-suspended at an OD600 of 0.2 in TE and 37% formaldehyde was added to a final concentration of 1% to cross-link proteins in the cell. Cells were incubated at room temperature for 30 min and subsequently for another 30 min on ice (Umbarger et al., 2011). The formaldehyde was quenched by adding glycine to a final concentration of 0.125 M and incubated on ice for 10 min. After centrifugation, cells were re-suspended in TE and lysozyme digestion was carried out as described to release the protein-DNA complexes (Umbarger et al., 2011). The samples were centrifuged and re-suspended in Hi-C lysis buffer and incubated on ice for 15 min (Lieberman-Aiden et al., 2009). From this step on the original Hi-C protocol employing HindIII (Lieberman-Aiden et al., 2009) was applied with some modifications. To summarize the Lieberman-Aiden et al. (2009) protocol, DNA in the cross-linked protein complexes is digested with HindIII endonuclease following cell lysis and free DNA ends are tagged with biotin. Blunt-ended DNA fragments are ligated under highly dilute conditions, resulting in preferential ligation of fragments that are within the same cross-linked DNA/protein complex. Next, crosslinks are removed, DNA is purified, biotin is eliminated from un-ligated ends, DNA is size-selected, and ligation products are selected for through a biotin pull-down. One modification we made to the published procedure was to reduce the concentration of biotin-14-dCTP (Life Technologies) by half. Also, ligation to Illumina-compatible sequencing adapters (Bioo Scientific) was carried out in solution before capture with streptavidin beads. The DNA sample was size selected after end-repair and before adapter ligation by gel extraction for fragment sizes ranging from 280 to 420 bp. The bead-captured Hi-C library was amplified by 10 cycles of PCR before a final cleanup with Ampure XP beads (Agencourt). The library was sequenced in a single run on an Illumina Miseq machine using 160 bp paired-end reads.

Sequence alignment and quality filtering

Reference assembly sequences were obtained from the NCBI RefSeq database (Pruitt et al., 2012) with the following accession numbers for each of P. pentosaceus (NC_008525), L. brevis (NC_008497, NC_008498, NC_008499), E. coli BL21 (NC_012892), E. coli K12 DH10B (NC_010473), and B. thailandensis (NC_007651, NC_007650). These sequences were pooled into a single reference database for sequence alignment. No quality filtering was performed on raw reads, leaving this to be performed later using alignment quality scores. Split-read sequence alignment was performed (independently for each read in a pair) using BWA MEM (Li, 2013; default parameters) against the pooled reference assemblies as well as (separately) against the metagenome assembly described above. Heat map visualizations and insert distribution plots were generated from unfiltered alignments using custom R (The R Core Development Team, 2010) scripts (see http://github.com/cb01/proxmine).

We investigated the effect of various alignment filtering parameters on the subsequent variant graph analysis (illustration of the concept of a variant graph can be found in Fig. S7). To this end, alignments of Hi-C reads to the reference genomes were filtered according to 24 parameter combinations, with three minimum mapping quality (0, 20, 60), two CIGAR filtering (none, CIGAR = 160 M), and four minimum insert filtering (0, 1 kb, 10 kb, 40 kb) conditions. In the latter, Hi-C read pairs were excluded when their alignments within the reference assembly had an insert size below the specified minimum, including cases of alignments spanning the linearization points of these assemblies (e.g., for E. coli, near coordinates 0 and 4686137). An open-source graph visualization tool, Gephi (0.8.2-beta), was used to visualize the Hi-C contig association network (Bastian, Heymann & Jacomy, 2009).

Contig clustering

We inferred grouping of metagenome assembly contigs by applying the Markov Clustering Algorithm (MCL) to a matrix of contig association data (van Dongen, 2000). MCL is an unsupervised clustering algorithm which simulates flow and accumulation of edge weights within a given weighted graph structure. It has a computational complexity of O(Nk2) given an implementation designed for sparse matrices. The matrix of edge weights provided to this algorithm was computed from contig association counts by normalizing edge weights according to the following formula, which corrects for the expected inflation of association between large contigs. Specified formally, let L be a set of contig lengths with member li denoting the length of contig i. Given a contig pair {i, j}, let cij denote the number of Hi-C read pairs with one end aligning in contig i and the other end in contig j. This count was normalized by the ratio of the square of the maximum contig length and the lengths li and lj of contigs i and j, respectively. cij′=maxL2ci,jlilj.

Prior to normalization, we filtered the contig association data for (1) contig associations greater than some minimum k, and (2) associations between contigs of size greater than L. We explored the (k, L) parameter space by performing normalization and MCL clustering for 205 parameter combinations, with 41 contig size minimums chosen evenly across [0, 40000] and five contact minimums, {0, 3, 5, 7, 9} . For each of the 205 filtering parameter combinations, clustering was performed using 100 different MCL inflation values chosen to span the interval [1, 2] in increments of 0.01.

Assessment of clustering quality

Each metagenome assembly contig was aligned to the reference assemblies to determine its species or strain of origin, allowing us to determine which strains were present in each cluster of contigs. This was done by extracting every 70 bp substring of the available contig sequences and aligning each of those back to the reference assemblies with BWA-MEM (default parameters, MapQ > 20). A contig was designated as originating from the species to which the greatest number of these substrings aligned. A measure of clustering quality was computed by sampling random pairs of contigs (N = 100,000 pairs) and comparing their cluster assignments to their species of origin. This random sampling was weighted according to the size of each contig such that the probability of sampling any contig from the set was equal to the size of that contig divided by the sum total of all contig sizes. Any contig not present in the clustering solution (but which was present as input for the clustering run that generated that solution) was added to the solution in a singleton cluster. For each sampled contig pair, if those contigs belonged to the same species and had been placed in the same cluster, they were counted as a true positive (“TP”). If they originated from the same species but had been placed in different clusters, they were counted as a false negative (“FN”). Likewise, contigs originating from different species that were placed in the same or different clusters were counted as false positives (“FP”) and true negatives (“TN”), respectively. The true positive rate (a.k.a. sensitivity, recall), false positive rate, positive predictive value (a.k.a. precision), and negative predictive value were calculated from these counts according to standard formulae. All clustering quality measures were computed in two ways, one treating the two E. coli strains as independent classes (strain-level) and another treating them as the same (species-level).

Analysis of SNP graph connectivity

A SNP graph is an undirected graph wherein SNP sites are nodes and edges link pairs of SNP sites that were observed together in a sequence read pair. We expected SNP graphs that were constructed using Hi-C data to be more densely connected than SNP graphs that were constructed using mate-pair data because Hi-C read pairs can span entire chromosomes while reads from mate-pair libraries span no more than 40 kb ± 5 kb. In this way, Hi-C provides global information while that provided by mate pairs is locally constrained. To quantify this, SNP graphs were constructed for the Hi-C data and simulated mate-pair data and the shortest path between randomly chosen SNP pairs was plotted relative to the distance between the those variants within the reference assembly. These graphs were constructed from alignments of reads to the E. coli K12 reference assembly after it had been masked at variant positions identified through pairwise sequence alignment of E. coli K12 and BL21 using progressiveMauve (Darling, Mau & Perna, 2010). In this way, the masked K12 reference assembly was used as a scaffold for our analysis of Hi-C and mate-pair variant graphs.

Hi-C read datasets typically contain a mixture of reads derived from ligation and non-ligation products, the latter having short inserts. In our analysis of SNP graph connectivity we sought to understand the contribution of these non-ligation products on the connectivity gains seen with Hi-C reads over mate-pairs. For comparison to our Hi-C reads, mate-pair read sets were computationally simulated for a range of sizes (5 kb, 10 kb, 20 kb, 40 kb). These were compared to three Hi-C read sets: the entire Hi-C dataset, Hi-C reads with inserts below 1 kb, and Hi-C reads with inserts above 1 kb.

SNP graph connectivity was analyzed using the simulated mate-pair read sets described above combined with Hi-C reads aligned and filtered for alignment qualities above or equal to 60 and for CIGAR encodings of 160 M Read pairs with both ends aligning to SNP positions were identified and for each corresponding SNP pair an edge was formed. Shortest path lengths between sampled SNP positions were computed using a custom breadth-first search program, relying on the Boost Graph Library (http://boost.org). The program constructs a graph from a user-specified SNP edge list and performs a breadth-first search to identify the shortest path length between a user-specified number of randomly selected SNP pairs. We calculated path lengths between 10,000 randomly chosen SNP pairs. To aid in visualization, the full range of variant separation distances was divided into 20 kb segments and the average path length was computed for each segment. These data were smoothed using locally-weighted scatterplot smoothing (LOWESS).

Results

The synthetic microbial community for metagenomic Hi-C

Five microorganisms were chosen to test the metagenomic Hi-C approach: Lactobacillus brevis, Pediococcus pentosaceus, Burkholderia thailandensis, Escherichia coli K12 DH10B and E. coli BL21 (DE3). These were selected because high quality reference genomes are available. In addition, the multiple replicons of B. thailandensis and plasmids present in L. brevis allowed us to explore whether Hi-C might link separate replicons present in the same cell. We selected two strains of E. coli (K12 and BL21) to evaluate whether Hi-C sequence data could be used to resolve inter-strain differences. Genome alignment of these two E. coli shows that 87.9% of their genomes can be aligned and that the average nucleotide identity across aligned regions is 99.5%. Finally, because differences in cell membrane structure and GC content could potentially lead to bias in DNA extraction and/or cross-linking efficiency, we selected two lactic acid bacteria (P. pentosaceus and L. brevis), which are low-GC, Gram-positive organisms for which only 39.5% of their genomes can be aligned with 84.3% average nucleotide identity across aligned regions.

Metagenome assembly

We generated a metagenome assembly of the synthetic microbial community that we could use as input for our analysis of the utility of Hi-C for species clustering. Hi-C sequencing data is biased by the distribution of restriction sites for the restriction enzyme used to construct the library as well as by other factors including GC content, restriction fragment length, and “mappability” (Yaffe & Tanay, 2011). Hi-C data contain numerous chimeric sequences and thus are not suitable for de novo contig assembly. Therefore, we simulated and assembled Illumina metagenomic sequence data at varying coverage levels to yield the assemblies summarized in (Table S1). The size (bp) of each of these assemblies was approximately 77% of the sum of the synthetic community reference genome sizes and this fraction did not change when increasing the quantity of input reads from 5x to 100x. Alignment of assembled contigs to the collection of reference genomes indicates that similar regions of the two E. coli genomes were co-assembled into single contigs (data not shown). The assembly on the lowest amount of input sequence (5x coverage) contained two misassembled contigs and three misassembled scaffolds. Assemblies at 50x and 100x coverage were free from misassembled contigs and scaffolds and were similar in terms of their contig counts, N50s, and total amounts of sequence assembled. We used the 100x coverage assembly (SOAP-3) for all further analysis.

Hi-C library statistics

A total of 20,623,187 read pairs were obtained from a single MiSeq run to yield ∼6.4 Gb of raw sequence data. Of these, 98.25% could be aligned back to the reference genomes by BWA MEM. A total of 21,260,753 (51.55% of original and 52.46% of raw aligned reads) read pairs were retained after filtering for both reads in the pair aligning at high quality (MapQ >= 60) and in full (CIGAR = 160 M). Due to either self-ligation or imperfect enrichment for ligation junctions, most of the reads present in the dataset represent local genomic DNA fragments. Therefore we classify read pairs mapping within 1,000 nt as fragment reads, while all other reads are considered to be Hi-C reads (3% of reads). The abundance of each replicon was estimated using filtered alignments and unfiltered alignments (Table 1), as well as by normalizing each unfiltered alignment count with the restriction site counts for each replicon (Table S2). These figures can only be used as approximate measures of abundance because these values are affected by the frequency of restriction sites in each organism and a multitude of other confounding factors (Morgan, Darling & Eisen, 2010). Insert distances derived from the alignment of Hi-C reads to the E. coli K12 genome were distributed in a similar manner as previously reported (Fig. 1; (Lieberman-Aiden et al., 2009)). We observed a minor depletion of alignments spanning the linearization point of the E. coli K12 assembly (e.g., near coordinates 0 and 4686137) due to edge effects induced by BWA treating the sequence as a linear chromosome rather than circular.

Table 1 Species alignment fractions.

The number of reads aligning to each replicon present in the synthetic microbial community are shown before and after filtering, along with the percent of total constituted by each species. The GC content (“GC”) and restriction site counts (“#R.S.”) of each replicon, species, and strain are shown. Bur1: B. thailandensis chromosome 1. Bur2: B. thailandensis chromosome 2. Lac0: L. brevis chromosome, Lac1: L. brevis plasmid 1, Lac2: L. brevis plasmid 2, Ped: P. pentosaceus, K12: E. coli K12 DH10B, BL21: E. coli BL21. An expanded version of this table can be found in Table S2.

Sequence	Alignment	% of Total	Filtered	% of aligned	Length	GC	#R.S.	
Lac0	10,603,204	26.17%	10,269,562	96.85%	2,291,220	0.462	629	
Lac1	145,718	0.36%	145,478	99.84%	13,413	0.386	3	
Lac2	691,723	1.71%	665,825	96.26%	35,595	0.385	16	
Lac	11,440,645	28.23%	11,080,865	96.86%	2,340,228	0.46	648	
Ped	2,084,595	5.14%	2,022,870	97.04%	1,832,387	0.373	863	
BL21	12,882,177	31.79%	2,676,458	20.78%	4,558,953	0.508	508	
K12	9,693,726	23.92%	1,218,281	12.57%	4,686,137	0.507	568	
E. coli	22,575,903	55.71%	3,894,739	17.25%	9,245,090	0.51	1076	
Bur1	1,886,054	4.65%	1,797,745	95.32%	2,914,771	0.68	144	
Bur2	2,536,569	6.26%	2,464,534	97.16%	3,809,201	0.672	225	
Bur	4,422,623	10.91%	4,262,279	96.37%	6,723,972	0.68	369	

Figure 1 Hi-C insert distribution.

The distribution of genomic distances between Hi-C read pairs is shown for read pairs mapping to each chromosome. For each read pair the minimum path length on the circular chromosome was calculated and read pairs separated by less than 1000 bp were discarded. The 2.5 Mb range was divided into 100 bins of equal size and the number of read pairs in each bin was recorded for each chromosome. Bin values for each chromosome were normalized to sum to 1 and plotted.

Clustering contigs by species with Hi-C

The experimentally derived Hi-C read pairs have a long tail in their insert distribution (Fig. 1; Figs. S3–S6) indicating that they provide information that can be used to link metagenome assembly contigs originating from distant parts of the same chromosome. We evaluated whether Hi-C reads could be used to group the simulated assembly contigs described above into clusters that correspond with each species’ genome. We tested this process on the SOAP-3 assembly, using only contigs with a length of at least 5 kbp. This threshold was applied to exclude short contigs that may not have a HindIII restriction site. As HindIII recognizes a 6 bp motif, it cuts on average every 4,096 bp. We note that >25% of a 4 Mbp genome is expected to have inter-site distances >10 Kbp in simulations that treat 6-cutter restriction sites as uniformly distributed (data not shown) suggesting that many of the contigs <10 Kbp may cluster poorly due to lack of restriction sites. The dataset was further reduced to exclude links among contig pairs which are associated by 5 or fewer read pairs.

We tested Markov clustering (van Dongen, 2000) on these data over a range of inflation (affecting cluster solution granularity) parameters (Table 2). In the best case, Markov clustering produced four clusters, each of which correspond to the nearly complete genome of a species in our synthetic community. In this clustering, the two strains of E. coli appear in the same group. When using the default inflation parameter of 2.0 we find that the data is under-clustered, but there are no false positive associations among contigs for this choice of input.

Table 2 Markov clustering of metagenome assembly contigs using Hi-C data.

A range of inflation parameters were applied, and the precision and recall for the resulting clusters was calculated as described in the text. An inflation parameter of 1.1 produced a near perfect clustering of contigs by species.

Inflation	Precision	Recall	# clusters	
2.0	1	0.19	33	
1.3	1	0.33	25	
1.125	1	0.98	5	
1.1	0.96	0.98	4	

To further understand the sensitivity of MCL to choices of filtering and inflation parameters, we performed clustering across the 204 filtering and 100 inflation (total of 20,400) conditions (see Methods: Contig Clustering) using MCL. A representative subset of all parameter combinations tested is shown in Fig. S2. These data suggest that once sufficient contact and contig size minimums have been applied, cluster solutions vary primarily in terms of their granularity (as the inflation parameter varies), not their PPV (remaining close to 1) or FPR (remaining close to 0). Low inflation values, close to 1, give clustering solutions with the highest TPR’s, but this does not hold true without sufficient filtering.

Association of species with metagenomic Hi-C data

We next sought to quantify the cellular co-localization signal underlying the above-described species clustering. For this analysis we studied Hi-C reads aligned directly to the reference assemblies of the members of our synthetic microbial community with the same alignment parameters as were used in the top ranked clustering (described above). We first counted the number of Hi-C reads associating each reference assembly replicon (Fig. 2; Table S3), observing that Hi-C data associated replicons within the same species (cell) orders of magnitude more frequently than it associated replicons from different species. The rate of within-species association was 98.8% when ignoring read pairs mapping less than 1,000 bp apart. Including read pairs <1,000 bp inflated this figure to 99.97%. Fig. 3 illustrates this by visualizing the graph of contigs and their associations. Similarly, for the two E. coli strains (K12, BL21) we observed the rate of within-strain association to be 96.36%. When evaluated on genes unique to each strain (where read mapping to each strain would be unambiguous), the self-association rate was observed to be >99%.

Figure 2 Metagenomic Hi-C associations.

The log-scaled, normalized number of Hi-C read pairs associating each genomic replicon in the synthetic community is shown as a heat map (see color scale, blue to yellow: low to high normalized, log scaled association rates). Bur1: B. thailandensis chromosome 1. Bur2: B. thailandensis chromosome 2. Lac0: L. brevis chromosome, Lac1: L. brevis plasmid 1, Lac2: L. brevis plasmid 2, Ped: P. pentosaceus, K12: E. coli K12 DH10B, BL21: E. coli BL21.

We observed that the rate of association of L. brevis plasmids 1 and 2 with each other and with the L. brevis chromosome was at least 100-fold higher than with the other constituents of the synthetic community (Fig. 2). Chromosome and plasmid Hi-C contact maps show that the plasmids associate with sequences throughout the L. brevis chromosome (Fig. 4; Figs. S3–S5) and exhibit the expected enrichment near restriction sites. This demonstrates that metagenomic Hi-C can be used to associate plasmids to specific strains in microbial communities as well as to determine cell co-localization of plasmids with one another.

Figure 3 Contigs associated by Hi-C reads.

A graph is drawn with nodes depicting contigs and edges depicting associations between contigs as indicated by aligned Hi-C read pairs, with the count thereof depicted by the weight of edges. Nodes are colored to reflect the species to which they belong (see legend) with node size reflecting contig size. Contigs below 5 kb and edges with weights less than 5 were excluded. Contig associations were normalized for variation in contig size.

Figure 4 Hi-C contact maps for replicons of Lactobacillus brevis.

Contact maps show the number of Hi-C read pairs associating each region of the L. brevis genome. The L. brevis chromosome (Lac0, (A), Spearman rank correlation) and plasmids (Lac1, (B); Lac2, (C)) show enrichment for local associations (bright diagonal band). Interactions between Lac1 and Lac0 (D) and Lac2 and Lac0 (E) are shown. All except Lac0 are log-scaled. Circularity of Lac0 became apparent after transforming data with the Spearman rank correlation (computed for each matrix element between the row and column sharing that element) in place of log transformation (A) indicated by the high number of contacts between the ends of the sequence. In all plots, pixels are sized to represent interactions between blocks sized at 1% of the interacting genomes. The number of HindIII restriction sites in each region of sequence is shown as a histogram on the left and top of each panel.

Variant graph connectedness

Algorithms that reconstruct single-molecule genotypes from samples containing two or more closely-related strains or chromosomal haplotypes depend on reads or read pairs that indicate whether pairs of variants coexist in the same DNA molecule. Such algorithms typically represent the reads and variant sites as a variant graph wherein variant sites are represented as nodes, and sequence reads define edges between variant sites observed in the same read (or read pair). We reasoned that variant graphs constructed from Hi-C data would have much greater connectivity (where connectivity is defined as the mean path length between randomly sampled variant positions) than graphs constructed from mate-pair sequencing data, simply because Hi-C inserts span megabase distances. Such connectivity should, in theory, enable more accurate reconstruction of single-molecule genotypes from smaller amounts of data. Furthermore, by linking distant sites with fewer intermediate nodes in the graph, estimates of linkage disequilibrium at distant sites (from a mixed population) are likely to have greater precision.

To evaluate whether Hi-C produces more connected variant graphs we compared the connectivity of variant graphs constructed from Hi-C data to those constructed from simulated mate-pair data (with average inserts of 5 kb, 10 kb, 20 kb, and 40 kb). To exclude paired-end products from the analysis, Hi-C reads with inserts under 1 kb were excluded from the analysis. For each variant graph constructed from these inputs, 10,000 variant position pairs were sampled at random, with 94.75% and 100% of these pairs belonging to the same connected graph component of the Hi-C and 40 kb variant graphs, respectively. These rates fell to 6.21%, 16.6%, and 32.38% for the 5 kb, 10 kb, and 20 kb mate-pair variant graphs, respectively (Table 3).

Table 3 Variant graph statistics.

Connectivity statistics are shown for variant graphs constructed from various simulated mate-pair (# kb, MP) and Hi-C read datasets. Graph constructed from all Hi-C data are compared to those constructed using only Hi-C read pairs with inserts over 1 kb. The Hi-C variant graphs are highly connected in contrast to the mate-pair graphs that have both lower connectedness and lower rates of variants occurring in the same connected components.

	Num. reads	Max	Avg.	% Same c.c	
5 kb, MP	10,287,315	71	14.81	6.21	
10 kb, MP	7,681,515	96	24.45	16.6	
20 kb, MP	4,871,227	94	27.58	32.38	
40 kb, MP	4,257,896	111	37.19	100	
Hi-C (all)	16,429,505	10	5.11	97.77	
Hi-C (>1 kb)	111,525	11	5.47	94.75	

Across conditions, variant graphs differed in terms of their connectivity, with Hi-C graphs showing the greatest connectivity. Despite having simulated an equal number of reads for each mate-pair distance, the numbers of variant positions linked by such reads was different across conditions. We observed that the variant graph derived from Hi-C data (>1 kb inserts, no alignment filtering), despite having the lowest number of variant links, had the lowest mean and maximum path length (5.47, 11; Table 3). Path length was not correlated with distance within Hi-C variant graphs, in contrast to the mate-pair conditions (Fig. 5). The lengths of paths between variant pairs in the mate-pair graphs did increase with distance, reaching maximums of 71, 96, 94, and 111 in the 5 kb, 10 kb, 20 kb, and 40 kb cases, respectively. We further examined the effect of alignment quality and completeness filtering and observed that in the latter case such filtering vastly reduced the rate at which variant positions occur within the same connected graph component.

Figure 5 Relationship of distance to degree of separation in Hi-C and mate-pair variant graphs.

The length of paths between random pairs of SNP sites in a SNP graph constructed from both Hi-C and mate-pair libraries of varying sizes (left; 5 kb, 10 kb, 20 kb, 40 kb), smoothed using locally-weighted regression.

Discussion

This study demonstrates that Hi-C sequencing data provide valuable information for metagenome analyses that are not currently obtainable by other methods. By applying Hi-C to a synthetic microbial community we showed that genomic DNA was associated by Hi-C read pairs within strains orders of magnitude more frequently than between strains. Hi-C reads associated genomic regions at distances not achievable with mate-pair or long-read sequencing technologies. The long-range contiguity information provided by Hi-C reads enabled us to perform species-level clustering of metagenome assembly contigs with perfect precision and recall scores when the input had been filtered sufficiently. We performed an exploration of the clustering parameter space to understand the factors affecting clustering quality and identified a number of key filtering parameters. Optimal filtering involved retention of only contigs that are large enough to contain (or occur near) a HindIII restriction site and furthermore to remove low-frequency contig associations that constitute a form of “noise”. Additional work is needed to develop methods to identify and remove “noise” from Hi-C datasets. Lastly, we compared the connectivity of variant graphs constructed from mate-pair and Hi-C read datasets, observing much greater connectivity in the latter case, illustrating the global nature of the Hi-C signal.

We also observed orders of magnitude greater rates of association between plasmids and chromosomes of their hosts than between plasmids and the genomes of other species. Based on this observation, we believe Hi-C has the potential to be used to study horizontal gene transfer. Given a metagenome assembly, Hi-C provides a means to link plasmid sequences to chromosomes of the host strain, and may provide the means to detect cases where plasmids have been transferred among co-existing species of bacteria. We have thus far demonstrated that Hi-C provides a signal of cell co-localization for the two plasmids present within the L. brevis genome. Alternative methods do not allow identification of which cells in a microbial community harbor such mobile DNA elements. Hi-C data has the potential to help quantify the dynamics of horizontal gene transfer and help characterize the spread of antibiotic resistance and virulence factors. It remains to be determined whether this signal will be sufficient to localize small, low-copy, or highly variable mobile elements within the species that contain them.

The resolving power of Hi-C and related methods such as ChIA-PET (Fullwood et al., 2009) when applied to complex natural microbial communities is as of yet undetermined. In principle, as the number of species and genotypes in a community grows the amount of sequence data required to resolve species and strains also grows. This challenge is common to all metagenomic approaches and is not specific to the Hi-C method described. Improvements on metagenomic analysis of complex communities may require integration of Hi-C data with other information sources such as sequence composition, phylogeny, and measurements of abundance.

The problems of differentiating contigs originating from different species is similar to that of differentiating contigs originating from different chromosomes of the same species. Recently a study reported the use of Hi-C to perform genome scaffolding of several individual eukaryotic genomes, first by inferring chromosomal groupings of contigs and then ordering sequences along the chromosome (Burton et al., 2013). Markov Clustering of Hi-C association data may be used to cluster contigs into chromosomal groups without specifying the number of chromosomes a priori. This may be important for samples where the number of chromosomes (e.g., tumor samples), species, or species abundances (e.g., environmentally-isolated microbial communities) are not known.

Hi-C analysis can be applied to communities other than environmentally-isolated microbial communities, such as pools of BAC clones. Heterogeneous tumor populations are analogous in some ways to microbial communities and Hi-C may be applied to identify sub-populations therein. The problem of resolving the membership of variants in closely related strains (between different cells) shares some common features with the problem of differentiating closely related haplotypes within polyploid eukaryotic genomes (within the same cells). Recent work has demonstrated that Hi-C data can be used to phase haplotypes in a diploid organism (Selvaraj et al., 2013). Our analysis indicates that the average degree of separation between variants within a Hi-C variant graph is dramatically lower than that in mate-pair variant graphs. This is significant because as the degree of separation between distant graph regions grows, error is compounded and the reliability of inferences regarding the phase of these regions declines. Our analysis thus indicates that Hi-C data provide an informative signal for the analysis of haplotype and strain mixtures.

Supplemental Information

Figure S1 Illustration of the signal provided by Hi-C for metagenome binning

Two bacterial cells are illustrated, each containing a single circular chromosome. For one genomic region in each of the two species, examples of associations that are likely (green; red is “not likely”) to be derived from Hi-C are illustrated.

Click here for additional data file.

Figure S2 Visualization of the impact of parameter choice on the quality of clustering solutions

A small-multiples plot is showing 5 × 5 combinations of contact minimum (top to bottom; 0, 3, 5, 7, 9) and contig size minimum (left to right; 1,000, 8,000, 15,000, 22,000, 29,000) thresholds. For each parameter combination, line plots show the quality (y-axis) of clustering solutions performed for inflation values in the interval [1, 2]. The quality of clustering solutions is measured in terms their true-positive rate (red), false-positive rate (green), positive predictive value (blue), and negative predictive value (black) are shown.

Click here for additional data file.

Figure S3 Hi-C contact frequency within Lactobacillus brevis genome

Contact frequency is visualized as a heat map, after normalization and application of the spearman rank correlation (matrix elements are the spearman correlation of the row and column of which they are the intersection). Circularity is apparent in the elevated contact between either end of the reference assembly sequence.

Click here for additional data file.

Figure S4 Hi-C contact map for Lactobacillus brevis plasmid 1

Contact maps show the number of Hi-C read pairs associating each region of the L. brevis plasmid 1. Contact values are Spearman rank correlation transformed following normalization. Pixels are sized to represent interactions between blocks sized at 1% of the interacting sequence. A minimal signal of circularity is apparent with enrichment for contact between the minimum and maximum positions within the reference assembly.

Click here for additional data file.

Figure S5 Hi-C contact map for Lactobacillus brevis plasmid 2

Contact maps show the number of Hi-C read pairs associating each region of the L. brevis plasmid 2. Contact values are Spearman rank correlation transformed following normalization. Pixels are sized to represent interactions between blocks sized at 1% of the interacting sequence. A signal indicative of circularity is not apparent.

Click here for additional data file.

Figure S6 Hi-C contact frequency within P. pentosaceus genome

Contact frequency is visualized as a heat map, after normalization and application of the spearman rank correlation (matrix elements are the spearman correlation of the row and column of which they are the intersection). Circularity is apparent in the elevated contact between either end of the reference assembly sequence.

Click here for additional data file.

Figure S7 Variant graph illustration

Two examples of variant graphs (non-data illustration). Variant nodes (circles) are linked by edges (light grey lines) derived from read pair data with small and medium (Graph I) or small, medium, and large (Graph 2) inserts. A path between two nodes (start, end) is illustrated and this path is shorter in the graph representing the dataset that includes larger-insert reads.

Click here for additional data file.

Table S1 SOAPdenovo assembly results

Statistics are shown for three assemblies, including the simulated coverage and the number of contigs (and scaffolds) present in the assembly. Assembly quality is reflected in the count of misassembled contigs and scaffolds (“contig error” and “scaffold error”). The percent of the total reference sequence size constituted by each assembly is also shown.

Click here for additional data file.

Table S2 Species alignment fractions (expanded table)

The number of reads aligning to each replicon present in the synthetic microbial community are shown before and after alignment filtering, along with the percent of total constituted by each species. The GC content and restriction site (R.S.) counts of each replicon, species, and strain are shown. Total and fractional raw alignment counts adjusted by R.S. counts are also shown, constituting our best approximation of relative abundances of synthetic community members. Bur1: B. thailandensis chromosome 1. Bur2: B. thailandensis chromosome 2. Lac0: L. brevis chromosome, Lac1: L. brevis plasmid 1, Lac2: L. brevis plasmid 2, Ped: P. pentosaceus, K12: E. coli K12 DH10B, BL21: E. coli BL21.

Click here for additional data file.

Table S3 Raw metagenomic Hi-C association counts

The number of Hi-C read pairs associating each genomic replicon in the mock community is shown without normalization.

Click here for additional data file.

Table S4 Normalized association counts

Shown are the counts of Hi-C read pairs associating each pair of replicons included in the synthetic community, normalized as described in the methods.

Click here for additional data file.

Supplemental Information 1 Hi-C of Mixed Bacterial Cultures

Click here for additional data file.

We wish to acknowledge the substantial efforts of Michael Lewis, administrator of the UC Davis Genome Center Cluster Computing Resource, to provide and maintain the computing resources on which these analyses were performed. We would also like to acknowledge Matthew DeMaere (UTS) for his contribution to Fig. 3.

Additional Information and Declarations

Competing Interests

Author Contributions

DNA Deposition

Data Deposition

Jonathan Eisen is an Academic Editor for PeerJ. The authors declare there are no competing interests.

Christopher W. Beitel conceived and designed the experiments, performed the experiments, analyzed the data, wrote the paper, prepared figures and/or tables, reviewed drafts of the paper, and conceived the method.

Lutz Froenicke conceived and designed the experiments, performed the experiments, reviewed drafts of the paper, and prepared Hi-C libraries.

Jenna M. Lang conceived and designed the experiments, performed the experiments, contributed reagents/materials/analysis tools, reviewed drafts of the paper, and prepared the mixture.

Ian F. Korf conceived and designed the experiments, analyzed the data, and reviewed drafts of the paper.

Richard W. Michelmore and Jonathan A. Eisen conceived and designed the experiments, contributed reagents/materials/analysis tools, and reviewed drafts of the paper.

Aaron E. Darling conceived and designed the experiments, analyzed the data, wrote the paper, prepared figures and/or tables, reviewed drafts of the paper, and conceived the method.

The following information was supplied regarding the deposition of DNA sequences:

Sequence Read Archives submission SRX377733.

Analysis scripts can be found at https://github.com/cb01/proxmine. Simulated assemblies are available at

http://dx.doi.org/10.6084/m9.figshare.1004473,

http://dx.doi.org/10.6084/m9.figshare.1004472, and

http://dx.doi.org/10.6084/m9.figshare.1004471

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
