# Peer review of "Strain- and plasmid-level deconvolution of a synthetic metagenome by sequencing proximity ligation products"

_PeerJ, doi:10.7717/peerj.415_

## Round 0.1 · original submission · Minor Revisions

Both reviewers thought your paper provided a great study with a valid approach and solid conclusions. Both also provide a number of minor issues for you to consider in your revision. I concur with these assessments of your work and encourage you to incorporate their feedback in a revised draft.

·

Basic reporting

I have a few concerns about the methods reporting and availability, addressed in more detail in my general comments. Specifically, both a detailed experimental protocol and a detailed computational protocol should be made available.

Experimental design

No comments. Looks good.

Validity of the findings

No comments. Looks good.

Additional comments

In this paper, Beitel et al. tackle the challenging problem of
metagenome contig binning using a Hi-C approach. Metagenome binning
is a frustrating challenge that, when done computationally, usually
relies on assumptions that are provably false for complex communities;
an experimental approach that scales would be tremendously important
for understanding complex microbial communities genomically. This
Hi-C procedure seems likely to be that approach, although there are
still significant technical challenges ahead.

Briefly, the authors show that the Hi-C approach of cross-linking DNA
to protein via fixation (to stabilize long-range physical
associations), then digesting and ligating (to connect fragments that
are physically associated), and then sequencing those ligation
products (to measure those long range associations), works as a way to
bin metagenome contigs. They also show that strain variants can be
effectively haplotyped in the same way. (Figure 2 is simply amazing
and should be on the cover of some journal somewhere.)

Broadly speaking, this is but a first (significant) foray into the
problem. The authors analyze a synthetic community consisting of only
5 microbes, in equal ratio; it's great that it works as well as it
does, but it remains to be seen how well the Hi-C approach will apply
to communities with higher diversity and richness. The conclusions
are not overstated, though, and I think it's a well-balanced and
important paper.

Note that Hi-C is technically difficult and notoriously finicky; this,
together with the coverage challenges associated with sequencing
high-diversity metagenomes, is likely to be the main barrier to
adoption of this technique in the future.

My only significant complaint about the paper is that some of the methods
need to be explicated better. Details below.

Specific comments:

The word "simple" should probably be added to "synthetic metagenome
sample" in the abstract.

Lines 31-39, references to the claims in here would be welcome.

Lines 54-58, references would again be welcome.

Lines 109-110, details on how contigs were mapped back to the
reference assemblies should be provided; I know from experience
that this can be parameter sensitive.

Lines 111-140, a detailed protocol should be provided somewhere
(figshare?); this is going to be very important for others trying to
replicate.

Line 152, "Custom R scripts" should be made available in a public and
archival location. (figshare? github?)

I found the Contig clustering section nicely detailed, but I was unable
to determine exactly how c_{ij} is calculated. Note c_{i,j}, line 173.
I can infer what was done here but it would be nice to have it made
explicit. The section is nicely detailed otherwise.

Line 182, again, details needed.

Line 256, the assembly size didn't change when increasing the quantity
of input reads from 5x to 100x? That's extremely surprising, given
that 5x coverage should miss a significant amount of true sequence due
to sampling error And only 77% of the community reference was
recovered?! This sounds like a horrible assembly process and I'm
somewhat mystified as to what is going on. Nonetheless, it doesn't
need to be addressed since the authors used SOAP-3 for all further
analyses but I would suggest omitting discussion of SOAP-1 and SOAP-2
unless it serves some purpose that I've missed.

Line 269, "some", line 272 "3%". I would suggest "many" or "most" instead
of "some"!

Line 283/Fig 1 is very nice!

Lines 301-308 suggest to me that the contig clustering analysis was
well done and very robust.

Line 323-325, excellent.

One last comment -- I hate being That Reviewer, but our to-be-released
any-day-now PNAS paper on soil metagenome assembly provides a new
computational method of binning species that is fairly assumption free
(see Pell et al., 2012, for the basic idea of partitioning reads based
on assembly graph connectivity; and Howe et al., 2014, available sometime in
March '14). Might be worth a read.

Signed,

C. Titus Brown

·

Basic reporting

The article is clear, well-written and interesting. The format, flow and sections (intro, methods, results, discussion, etc) are appropriate and complete. The figures and tables are appropriate and generally high-quality

Experimental design

The authors present a very interesting study utilizing a Hi-C approach in a metagenomic context. They apply their approach using a synthetic (in vitro) bacterial community. They show that their method provides experimental insight that is not available from typical sequencing-based metagenomic studies such as a more accurate approach for strain genotyping. Their methods are well-described and complete.

Validity of the findings

They apply their approach using a synthetic (in vitro) bacterial community, and show that their method provides experimental insight that is not available from typical sequencing-based metagenomic studies such as a more accurate approach for strain genotyping. Based on their methods, experiments and computation, I believe their results are valid and of high quality.

Additional comments

Summary: The authors present a very interesting study utilizing a Hi-C approach in a metagenomic context. As the authors state, this type of approach provides information that is not available using a standard sequencing-based metagenomic approach. They apply their approach using a synthetic (in vitro) bacterial community. The show that their method provides experimental insight that is not available from typical sequencing-based metagenomic studies such as a more accurate approach for strain genotyping (when multiple strains of the same species are in the sample) and in identifying incidences of horizontal gene transfer

The article is clearly written and the experimental details are clear and complete. I have the following comments/concerns for this manuscript:

1. What is the importance of the simulated reads and assembly in this dataset? Would similar result be obtained by merely aligning the reads to the reference genomes? Could the method still be used if deep sequencing data is not available for a sample?

2. Even though the amount of co-localization of the L. brevis plasmids with other genomes in the synthetic community are low (compared to co-localization within the species), it still seems to show interaction with other contigs from other species. Is this likely due to experimental noise, or is this a real phenomenon? If noise, how can one distinguish between background/noise and real interactions or gene transfers?

3. In the discussion, the authors mention that their approach can be used to detect horizontal gene transfers. I agree that this is the perfect approach for doing this analysis. However, the authors do not demonstrate this result. This decreases the impact and my enthusiasm for the manuscript significantly. Although I do not believe this paper should be rejected without this important piece of data, I would strongly recommend that the authors demonstrate their approach’s ability to detect gene transfers in an experimental setting.

4. The authors provide the commands for generating the simulated sequencing data, but are these data available for download?

---

## Round 0.2 · accepted · Accept

Thank you for your effect the thorough revision addressing the previous reviewers' concerns. I feel your paper is now ready for publication.